# Patients’ Perception and Knowledge about Influenza and Pneumococcal Vaccination during the COVID-19 Pandemic: An Online Survey in Patients at Risk of Infections

**DOI:** 10.3390/vaccines9111372

**Published:** 2021-11-22

**Authors:** Paul Loubet, Jalini Rouvière, Adeline Merceron, Odile Launay, Albert Sotto

**Affiliations:** 1INSERM U1047, Department of Infectious and Tropical Diseases, CHU Nîmes, Université Montpellier, 30900 Nîmes, France; albert.sotto@chu-nimes.fr; 2Pfizer, 75668 Paris, France; jalini.rouviere@pfizer.com; 3IPSOS, 75013 Paris, France; adeline.merceron@ipsos.com; 4Faculté de Médecine Paris Descartes, Université de Paris, AP-PH, Inserm, CIC Cochin Pasteur, 75231 Paris, France; odile.launay@aphp.fr

**Keywords:** COVID-19 vaccines, seasonal influenza vaccines, immunization, immunocompromised, pneumococcal vaccines, vaccination, vaccination coverage

## Abstract

Introduction: The objective of our study was to assess, in an at-risk population, perception and knowledge about influenza and pneumococcal vaccinations. Methods: An anonymous web-based survey was submitted to patients recruited in France, from both an Ipsos internal panel and AVNIR patient associations. The study was conducted between July and October 2020, in the context of the COVID-19 pandemic. Results: Overall, 2177 questionnaires from patients at risk of infection were analyzed. Almost all respondents (86%, 1869/2177) declared themselves to be favorable to vaccination. Nearly half of the patients (49%, 1069/2177) were aware of which vaccine was recommended for their specific situation. This percentage was significantly (*p* < 0.001) higher for members of a patient association and for people affected by multiple chronic conditions and varied according to the type of condition. Almost two-thirds of patients (1373/2177) declared having been vaccinated during the 2019/2020 influenza season, and 41% (894/2177) were certain about being up to date with the pneumococcal vaccination. The main barriers to vaccination for influenza are the fear of side effects, doubt regarding the efficacy of the vaccine and for pneumococcal vaccination, and the absence of suggestions by the healthcare professionals (HCPs), as 64% of respondents were not recommended to obtain pneumococcal vaccination. To improve vaccine coverage, information is of prime importance and GPs are recognized as the main HCP to inform about vaccination. Nearly two-thirds (62%, 1360/2177) of patients declared that the COVID-19 pandemic convinced them to have all the recommended vaccines. Conclusion: Our study highlighted the nonoptimal vaccine coverage in at-risk populations despite a highly positive perception of vaccines and confirmed that physicians are on the front lines to suggest and recommend these vaccinations, especially in the current pandemic context, which may be used to promote other vaccines.

## 1. Introduction

There is a significant burden of influenza and pneumococcal community-acquired pneumonia in adults every year including the number of infections, general practitioners (GP) and emergency room consultations, and related and non-related hospitalizations and deaths [1,2]. The latter point being particularly important with influenza, which is responsible, in nearly half of cases, for hospitalization related to non-respiratory complications as suggested in a large cross-sectional study using data from the US Influenza Hospitalization Surveillance Network [3]. This burden is higher in the elderly and those with chronic conditions, especially in those who are immunocompromised, whose number is increasing and who are at higher risk of infection and/or developing a severe form of diseases [4,5,6].

Vaccination, which is the most effective strategy for preventing influenza, pneumococcal, and COVID-19 infections and their complications, is recommended in these at-risk patients [7,8].

Previous surveys have identified the French population’s reluctance regarding vaccination and low vaccination coverage, especially among at-risk patients, despite these recommendations [9,10,11].

Up-to-date information is needed to better understand people’s attitudes toward vaccines as well as to identify the factors that influence at-risk patients’ vaccination intention in the context of the COVID-19 pandemic in order to tailor messages about vaccination accordingly.

We performed an online survey of patients at risk of severe influenza and/or pneumococcal disease to assess their perception and knowledge about vaccination, particularly in the context of the COVID-19 pandemic.

## 2. Methods

This survey is a French prospective web-based survey carried out between 29 July and 12 October 2020, conducted by Ipsos, the world’s third largest market research company, part of Syntec and the ESOMAR European Society for Opinion and Market Research. This survey is the third survey initiated and conducted in France with the AVNIR (Associations VacciNation Immunodéprimées Réalité) group and supported by Pfizer.

The AVNIR group encompasses 12 nonprofit French patient associations supporting immunocompromised patients and those with chronic conditions. The group’s aim is to promote supportive care among these patients to decrease the burden of infections.

AVNIR launched the first online survey in 2013 to gather information on vaccine coverage, knowledge, and attitudes toward vaccination among its members [10].

### 2.1. Study Population and Data Collection

The studied population comprised patients at risk of severe influenza and/or pneumococcal disease(s). The patients were recruited on a volunteer basis from both an Ipsos internal panel and AVNIR patient associations, without any quotas between the samples.

The prerequisites for survey participation were that individuals needed to be at least 16 years old, for the patients selected through the Ipsos Access Panel Online, and to answer a preliminary question on which disease(s) they were suffering from (selected from a list of pathologies or health disorders). The sample was constituted by randomly drawing from the database of panelists who were eligible for the study.

The AVNIR group gathers data on patient associations for most immunodeficiency diseases and diseases that require immunodepressive treatments (autoimmune inflammatory rheumatic diseases, inflammatory bowel disease (IBD), psoriasis, systemic scleroderma, chronic heart and kidney failure (including patients who have gone through organ transplantation), leukemia, chronic myeloid leukemia, and lymphoma).

The survey was promoted through both Ipsos and the websites of each patient association. A reward system that involves accumulating points was part of a loyalty program developed by Ipsos for respondents; patients from associations did not benefit from the survey.

The anonymous questionnaire (see full questionnaire in Appendix A) provided the following information: sociodemographic data (age, gender, occupation, area of residence, number of persons in the household), membership or non-membership in patient associations, the type and duration of treatment received, and the number of medical appointments per year.

Some questions allowed us to assess patients’ level of knowledge about pneumococcal infection and their risk perception concerning influenza, pneumococcus, and COVID-19. Respondents were also asked to rate on a 4-point scale how favorable they were toward vaccination. Moreover, questions were used to investigate how knowledgeable they were about vaccination and the recommended vaccines in the specific case of their disease, which healthcare professional was the first to propose and follow vaccination recommendations, and how vaccine uptake is recorded. Respondents were instructed to answer whether they had received seasonal influenza vaccines during the previous season (2019/2020, influenza vaccination campaign started on 28 October 2019, before the beginning of the pandemic of the SARS-CoV-2) and whether their pneumococcal vaccination status had been updated. They also had to express their perspective on motivating factors and concerns about influenza and pneumococcal vaccines. Finally, they were asked to state what they believed to be on the most reliable information source and identify items on which they wished for more information to improve vaccination coverage.

### 2.2. Statistical Analysis

Only complete questionnaires were analyzed. Descriptive statistics were used to characterize the population, including proportions, means and standard deviations (SDs) for normally distributed variables, and medians and interquartile ranges (IQRs) for nonparametric data. Fisher’s exact test and the Wilcoxon rank sum test were used to compare characteristics, knowledge, and attitudes. Self-reported vaccine uptake is given with 95% confidence intervals (95% CIs). We tested sociodemographic variables, care modalities, knowledge, factors associated with vaccine uptake, and attitudes toward vaccination in a univariate analysis. All analyses were performed using COSI software.

### 2.3. Ethical Considerations

IPSOS represents and warrants, directly and/or through its subcontractors, and it has, as a data controller, done the following:Complied with all applicable laws and regulations relating to privacy, security, and data protection (including but not limited to EU Data Protection Regulation 2016/679; “GDPR” and e-privacy regulations);Determined the applicable legal ground to contact the eligible individuals and implemented appropriate security measures to protect the information, including determining an appropriate data retention period;Provided appropriate notice to and/or obtained the prior and explicit consent from the individuals to use their data for the provision of services and creation of the deliverable, including to participate in the survey.

Participants were thus asked to submit an online informed consent form for their participation in the study before accessing the questionnaire. Data were collected anonymously, and participants had the right to access their answers. Only the aggregate data were analyzed and are shown in the final report.

## 3. Results

### 3.1. Study Population

Overall, 2177 questionnaires from patients at risk of infection were analyzed: 800 patients recruited through the Ipsos Panel and 1377 from the AVNIR network. Among these respondents, 1350 were female (62%), 704 (32%) were >65 years old (mean age was 56.6 ± 14.5 years), and 62% were immunocompromised. The demographic characteristics and the breakdown according to their chronic condition are described in Table 1.

### 3.2. Perception of the Risk of Contracting Some Vaccine-Preventable Infections

Most patients were aware that if they had one of these infections, the consequences would be more severe than for the general population. This risk was especially perceived as higher for COVID-19 (81%) but less so for pneumococcal (76%) and influenza infections (74%). Younger people perceived less risk of any severe infection or complications than older people did. Among patients under 35 years old, only 43% declared that contracting COVID-19 would lead to severe complications, compared with 35% for pneumococcal infection and 28% for flu. Being a member of a patient association was also a determinant of proper risk evaluation.

Only 36% (774/2177) of the population declared knowing rather precisely what pneumococcal infection was; this rate increased to 49% among members of a patient association.

### 3.3. Knowledge and Attitudes toward Vaccination

#### 3.3.1. Knowledge

Almost all respondents (86%, 1869/2177) said they were favorable to vaccination. Nearly half of patients (49%, 1069/2177) were aware of which vaccine was recommended for their specific situation. This percentage was significantly higher for members of a patient association and for people affected by multiple chronic conditions. The rate of patients who declared that they were well informed about vaccine recommendations also varied according to their chronic condition (Table 2).

#### 3.3.2. Attitudes toward Vaccination

Nearly two-thirds of patients (1373/2177) declared that they were vaccinated during the 2019/2020 influenza season. Influenza vaccine coverage was significantly lower for young patients (under 35 years old) and higher for patients with multiple chronic diseases. The rate of influenza vaccination was significantly higher (81%) for patients who considered their risk of influenza infection more important than the general population. Referring physicians (GPs and specialists) were the main actors involved in the vaccine process: GPs recommended the influenza vaccine for 47% (650/1373) of respondents who received the vaccine, and specialists made recommendations for 42% of respondents (573/1373).

The two main reasons for not being vaccinated against influenza were the fear of side effects (31%) and doubt regarding the efficacy of the vaccine (26%). Of note, 23% of patients were not offered an influenza vaccination.

In our study, the four-choice question for measuring vaccine uptake was “Would you say you are updated with your pneumococcal vaccinations?”: 41% (894/2177) of respondents declared “Yes, I’m sure”; 24% (524/2177) declared “Yes, I believe so, but I’m not sure”; 16%, declared “No, I’m not updated”; and 19% declared “I don’t know”. The proportion of patients who were certain about being up to date was higher for members of a patient association (50%, *p* < 0.01) and in those with more than one pathology (47% vs. 33%, *p* < 0.001). The self-reported rate of pneumococcal vaccination coverage varied according to the pathology (Table 3).

Approximately 9 out of 10 patients with updated vaccine status indicated that vaccination was suggested by a general practitioner (GP) (48%, 676/1418) or by a specialist (48%, 687/1418).

The main barrier to pneumococcal vaccination for patients without updated vaccine status was the absence of vaccination suggestions (64%, 218/343).

#### 3.3.3. Impact of the COVID-19 Pandemic

Nearly two out of three (62%, 1360/2177) patients declared that the COVID-19 pandemic convinced them to have all the recommended vaccines. Nevertheless, the proportion of respondents stating that the pandemic did not convince them remained high: 30% (657/2177) were not convinced to accept the anti-pneumococcal vaccine, and 28% (608/2177) were not persuaded to receive the influenza vaccine.

### 3.4. Ways to Improve Vaccination Coverage

GPs, compared to specialist practitioners, nurses, or pharmacists, were recognized as the main type of healthcare professional to inform people about vaccination. Many patients did not receive any recommendation for vaccinations or any information about the consequences of these respiratory infections (Table 4).

Access to information about vaccination was considered easy by patients, but some specific points could be more systemically explained: the recommended vaccines according to the disease and the treatments, the level of reimbursement, and the best time to be vaccinated.

The patients were asked about the main measures that could incite them to be vaccinated: receiving information from the national health insurance system about the pneumococcal disease and vaccine at the same time that the annual reminder about the influenza vaccine was the most plebiscite strategy. Other measures were approved by more than 75% of respondents: being informed about vaccination systematically as soon as the physician announced the diagnosis of the chronic disease, obtaining an easy-to-access digital tool for recording vaccinations, and being able to receive the vaccines from physicians.

## 4. Discussion

Our study, conducted in France on patients with chronic diseases at risk of influenza and pneumococcal-related infections, aimed to measure patients’ perceptions, knowledge, and attitudes about influenza and pneumococcal vaccination. The study was set up during the COVID-19 pandemic and before the availability of COVID-19 vaccines; this made it possible to measure the impact of the pandemic on the vaccination intentions of these patients. Knowledge of the barriers and the associated factors is an important element that needs to be considered in developing specific communication strategies among specific target groups to increase vaccination coverage.

### 4.1. The Willingness to Be Vaccinated after the COVID-19 Pandemic

The risk of contracting an infection such as COVID-19, influenza, or a pneumococcal infection was perceived by most of our respondents to be greater than the risk faced by the general population. In our study, a very large majority (86%) of patients declared themselves in favor of vaccination. The COVID-19 pandemic convinced 62% of respondents to be vaccinated with all recommended vaccines.

However, it remains difficult to predict the impact of the COVID-19 pandemic and mass vaccination against SARS-CoV-2 on the opinion of the general population and at-risk patients on vaccination.

### 4.2. The Vaccine Uptake in France and Europe and Associated Factors

In France, vaccination policies are issued by the Ministry of Health. Influenza vaccination is recommended annually for people 65 years of age and older and for people at risk of severe or complicated influenza [12].

Every year, most of the people targeted by these recommendations receive a voucher from the French health insurance system inviting them to be vaccinated against influenza. The vaccine is reimbursed; it is purchased at the pharmacy and can be administered by various health professionals including physicians, midwives or nurses, and recently pharmacists. Pneumococcal vaccination is recommended only for at-risk adults independently of age. French recommendations were updated in 2013 and 2017, and vaccination with both PCV13 and PPSV23 is currently on the admitted schedule.

In our study, 63% reported being vaccinated against influenza, a nonoptimal rate, while three-quarters (1642/2177) of patients admitted that vaccination was recommended to them.

This percentage matches the official French figures provided by the national health insurance system [13]: for the 2020–2021 season, among patients over 65 years old, the rate of vaccination was 59.90%, which represents a very large increase compared to 52% in the previous year (2019–2020 season). Among patients under 65 years old with comorbidities, the rate of vaccination was 38.70% (compared with 31% in 2019–2020). This better coverage is probably linked to the COVID-19 pandemic in 2020 and greater risk awareness. Robert J. et al. [14] analyzed influenza vaccine coverage from the French EGB database, a permanent sample of all individuals benefiting from social security in the general population. For those aged 65 and over with a potential risk, this coverage was estimated at 42.7% for 2014–2015: this figure corroborates the increase observed.

The self-reported data in our study are probably overestimated. Several years ago in France, in 2013 and 2016, two similar web-based surveys were conducted through the same French patient associations. In 2013, vaccine uptake by patients with an increased risk of severe influenza was 59% (95% CI (57–60) [3]). In 2016, among French patients with IBD, the rate was lower, with 34% [4] of patients with IBD being younger than our population, explaining the higher coverage in our survey.

In a literature review conducted in 2016 to evaluate the efficacy and coverage of influenza vaccines for at-risk patients [15], the coverage rate varied according to the subgroups at risk, from 59% for solid organ transplant recipients to 67% for those with chronic kidney disease. Recent reports from the ECDC in seven European countries in the seasons 2015/16, 2016/17, and 2017/18 and in Germany, over the course of 10 years, showed similar immunization rates in at-risk patients [16,17].

The pneumococcal vaccine uptake rate was around 41% in our study and was similar to the rates found in the two previous AVNIR studies [10]. In all the studies, recommendation of the vaccine by a healthcare professional appears to be an important factor associated with vaccination uptake.

As for the influenza vaccine, pneumococcal vaccine uptake was overestimated in our study compared with the findings of a large national survey conducted in France using national panels of medical records of 2000 GPs and 1000 specialist practitioners in ambulatory care (4%) [9], with nearly 100,000 individuals with chronic conditions in UK (32%) [18] or with nearly 200,000 immunocompromised in Germany (2% to 7%) [19].

In addition to vaccination recommendation by a health care professional (HCP), our study identified other factors associated with the vaccination: age, number of chronic conditions, and membership in a patient association. Some groups of patients have higher vaccine coverage according to their pathologies, such as patients with congenital abnormalities, those with respiratory diseases, and transplant patients.

Our results are consistent with those described in the literature. Age and number of chronic conditions were positively associated with vaccination against influenza and pneumococcal diseases in Belgium [20] and Spain [21] in at-risk groups. Similarly, patients with COPD, HIV, and solid organ transplantation (SOT) were more likely to be vaccinated against influenza and pneumococcal diseases than patients with heart failure. Confidence in the importance of vaccines (rather than in their safety or effectiveness) had the strongest univariate association with vaccine uptake in the large study of global vaccine confidence performed between 2015 and 2019 across 149 countries [22]. The determinants of vaccine uptake across the globe show strong consistency, with decreased uptake associated with being male or having fewer years of education.

### 4.3. Barriers and Ways to Improve Vaccine Coverage

In our study, we noticed some differences between influenza and pneumococcal vaccination barriers. The fear of side effects and doubts about effectiveness are the main reasons given by patients who refuse the flu vaccine. A smaller number of patients declared they did not perceive themselves to be at risk of influenza infection. The same concerns have been described in a prior literature review [23]. On the other hand, the fact that patients are not aware of the recommendation is a major factor that explains the absence of vaccination, especially for pneumococcal vaccination [22]. The literature review performed by Doornekamp et al. [24] highlighted the fact that immunocompromised patients were often unaware of the recommended vaccinations, and they demonstrated the strong correlation between physicians’ recommendations and higher uptake. Additionally, in our study, 64% of respondents were not recommended to obtain pneumococcal vaccination.

These results emphasize patients’ concerns regarding vaccination and provide some clues about key messages to be disseminated. Patients need to be reassured about vaccine safety and informed about risk factors for having complications related to respiratory infections such as flu and pneumococcal infections in the same way that has been done with COVID-19.

Concerns on reduced and/or short-term vaccine efficacy, especially influenza vaccines [25,26], might play an important role in patients reluctancy to get vaccinated, especially in younger ones who feel less at risk of severe diseases. This point underlines the need for clear explanation on the benefits/risk ratio in these patients, especially in the immunocompromised.

Vaccine recommendations are an important lever, and they are better known for influenza than for pneumococcus. Our study confirmed that physicians (GPs or specialists) are on the front lines to suggest and recommend these vaccinations. Furthermore, patients declared that they would be motivated to be vaccinated if their physician was vaccinated; therefore, the vaccination of healthcare workers is not only an option to protect patients, but also a way to serve as an example; this is key in promoting vaccine acceptance [27]. Berrada et al. [28] defined three main themes in their qualitative study concerning vaccination hesitancy: restoration of trust in vaccine policy, improvement of the initial and further training of health care workers, and better communication with the population.

The sample of the population in our study was not weighted by disease to correct for over- or underrepresentation and reduce the bias of nonrepresentation of some pathologies. In our sample, for instance, patients with psoriasis are overrepresented, yet only patients with severe psoriasis (treated by immunosuppressive therapy or biotherapy), who represent less than 40% of psoriasis patients, can be considered at risk of infectious disease. Concomitantly, patients with diabetes are underrepresented at only 12% of our sample. Moreover, web-based surveys are intended for individuals who are comfortable with the internet; thus, seniors and precarious people are likely underrepresented. In addition, self-reported data may introduce memorization or social desirability biases.

## 5. Conclusions

Our study highlights the low vaccine coverage in at-risk populations despite a highly positive perception of vaccines. The COVID-19 pandemic undoubtedly offers an opportunity to educate patients about the risks of infectious complications and to increase vaccine coverage. Initiatives to increase vaccine recommendation awareness among healthcare professionals and at-risk patients are still necessary.

## Figures and Tables

**Table 1 vaccines-09-01372-t001:** Patient characteristics.

Total Population	*N* = 2177
Gender	
Female *n* (%)	1350 (62.01%)
Age	
Age mean years (Standard Deviation)	56.6 (14.5)
Patients from 16 y to 35 y, *n* (%)	204 (9%)
Patients from 36 y to 50 y, *n* (%)	513 (24%)
Patients from 51 y to 65 y, *n* (%)	756 (35%)
Patients over 65 y, *n* (%)	704 (32%)
Membership in a patient association	838 (38%)
Number of pathologies	
Patients with only one pathology, *n* (%)	975 (45%)
Patients with more than one pathology, *n* (%)	1202 (55%)
Chronic conditions	
Inflammatory rheumatic diseases	628 (29%)
Rheumatoid arthritis, *n* (%)	289 (13%)
Psoriatic arthritis, *n* (%)	251 (12%)
Spondyloarthritis, *n* (%)	122 (6%)
Systemic scleroderma, *n* (%)	32 (2%)
Drepanocytosis, *n* (%)	8 (0%)
Inflammatory bowel diseases (IBD)	83 (4%)
Inflammatory skin diseases	585 (27%)
Psoriasis, *n* (%)	570 (26%)
Lupus, *n* (%)	19 (1%)
Autoimmune diseases, *n* (%)	462 (21%)
Treated with immunosuppressive therapy including biologics and/or systemic corticoids, *n* (%)	279 (13%)
Other autoimmune diseases, *n* (%)	225 (10%)
Respiratory diseases, *n* (%)	433 (20%)
Chronic obstructive pneumopathy disease (COPD), *n* (%)	279 (13%)
Severe asthma, *n* (%)	153 (7%)
Chronic respiratory failure, *n* (%)	124 (6%)
Emphysema, *n* (%)	107 (5%)
Cardiovascular diseases, *n* (%)	409 (18.78%)
Pulmonary hypertension, *n* (%)	335 (15%)
Chronic heart failure (CHF), *n* (%)	100 (5%)
Diabetes needing treatment, *n* (%)	258 (12%)
Kidney diseases, *n* (%)	195 (9%)
Chronic kidney failure (transplanted), *n* (%)	111 (5%)
Chronic kidney failure before supplementation, *n* (%)	56 (3%)
Chronic kidney failure (dialysis), *n* (%)	45 (2%)
Nephrotic syndrome, *n* (%)	17 (1%)
Tx patients, *n* (%)	181 (8%)
Solid organ transplant (SOT), *n* (%)	156 (7%)
Donor organ, *n* (%)	85 (4%)
Hematopoietic stem cell transplanted, *n* (%)	11 (0.5%)
Neoplasia (solid organ and malignant blood diseases)	152 (7%)
Patients living with infectious disease (HIV, chronic hepatitis)	35 (2%)
Congenital, chromosome abnormalities	47 (2%)
Congenital heart disease	42 (2%)
Asplenia	7 (0%)
PID Primary immunodeficiency	14 (1%)
Treatments	
Biologics	321 (15%)
Systemic corticoids	323 (15%)
Chemotherapy	89 (4%)
Immunosuppressive therapy (transplant rejection and other immunosuppressive)	524 (24%)

**Table 2 vaccines-09-01372-t002:** Vaccine recommendation awareness.

Patients Who Declared Being Well Informed about Vaccine Recommendations		
Total population (*N* = 2177)	1069/2177 (49%)	
Membership in a patient association (*N* = 838)	561 (67%)	*p* < 0.001 *
People affected by more than one chronic condition *(N* = 1202)	721 (60%)	*p* < 0.001 *
According to the chronic condition	*N* (%)	
Transplanted (Tx) (*N* = 181)	125 (69%)	
Inflammatory skin diseases (*N* = 585)	199 (34%)	
Autoimmune diseases (*N* = 462)	268 (58%)	
Cardiovascular diseases (*N* = 409)	168 (41%)	
Inflammatory bowel diseases (IBD) *(N* = 83)	38 (46%)	
Kidney diseases (*N* = 195)	115 (59%)	
Respiratory diseases (*N* = 433)	299 (69%)	
Inflammatory rheumatic diseases (*N* = 628)	383 (61%)	
Diabetes (*N* = 258)	103 (40%)	
Patients living with infectious diseases (*N* = 35)	16 (46%)	
Congenital/chromosome abnormalities (*N* = 47)	33 (70%)	
Neoplasia (solid organ and malignant blood diseases) (*N* = 152)	67 (44%)	

* Fisher’s exact test.

**Table 3 vaccines-09-01372-t003:** Pneumococcal vaccination coverage.

Respondents “Sure about Being Up to Date on Pneumococcal Vaccinations”	*N* (%)	Difference vs. Total Population
Total population (*N* = 2177)	894 (41%)	
Membership in a patient association (*N* = 838)	210 (50%)	*p* < 0.001
People affected by more than one chronic condition (*N* = 1202)	266 (47%)	*p* < 0.001
According to the chronic condition		
Congenital/chromosome abnormalities (*N* = 47)	28 (59%)	*p* < 0.001
Respiratory diseases (*N* = 433)	242 (56%)	*p* < 0.001
Inflammatory rheumatic diseases (*N* = 628)	339 (54%)	*p* < 0.001
Patients living with chronic infectious diseases (HIV, HCV) (*N* = 35)	19 (54%)	
Autoimmune diseases (*N* = 462)	226 (49%)	*p* < 0.001
Transplanted (Tx) (*N* = 181)	72 (40%)	
Cardiovascular diseases (*N* = 409)	160 (39%)	
Inflammatory bowel diseases (*N* = 83)	32 (39%)	
Kidney diseases (*N* = 195)	72 (37%)	
Neoplasia (solid organ and malignant blood diseases) (*N* = 152)	53 (35%)	
Inflammatory skin diseases (*N* = 585)	193 (33%)	*p* < 0.001
Diabetes (*N* = 258)	73 (28%)	*p* < 0.001

**Table 4 vaccines-09-01372-t004:** Information about vaccine recommendations. Topics addressed by patients’ healthcare professionals (HCPs) over the last 12 months.

Total Respondents *N* = 2177	
HCPs recommending flu vaccine	
General practitioner	1265 (58%)
Specialist	783 (36%)
Pharmacist	135 (6%)
Nurse	98 (5%)
None of them	562 (26%)
HCPs checking pneumococcal vaccine status	
General practitioner	1000 (46%)
Specialist	620 (28%)
Pharmacist	52 (2%)
Nurse	45 (2%)
None of them	820 (38%)
HCPs explaining flu and pneumococcal consequences	
General practitioner	1091 (50%)
Specialist	767 (35%)
Pharmacist	98 (5%)
Nurse	82 (4%)
None of them	712 (33%)

## Data Availability

Data was obtained from Ipsos and are available from the authors with the permission of Ipsos.

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
