# Peer review of "Patients’ Perception and Knowledge about Influenza and Pneumococcal Vaccination during the COVID-19 Pandemic: An Online Survey in Patients at Risk of Infections"

_vaccines, 2021, doi:10.3390/vaccines9111372_

Round 1

Reviewer 1 Report

The introduction and discussion parts need reshaping

The reluctance of people to allow vaccination should be explained.

In fact some publications which allow some uncertanity should be mentioned:

Heo JY Human Vaccines and immunotherapeutics 2018;14(3) 744-749,

Okoli GN,Open Forum Infectuous Diseases:Decline in seasonal Influenza Vaccine Effectivness....,Chow EJ,JAMA Network 2020 March 20.

Dänemark has vaccinated 90% of the population against COVID-19 but infection numbers are exploding!

It is also important to discuss why patients seek help in the emergency rooms and possibly die in the hospital as this offers one justification for vaccination

The study,supported by Pfizer, aimed to find out how many of the persons with different immunosuppression
were informed about vaccines against influenza,against pneumococcus and against SARS-CoV-2 and were vaccinated
.Although a quite high number of persons were well informed the vaccine coverage was quite low.
The topic could be relevant to understand why many immunosuppressed persons, although well informed about vaccines
do not get vaccinated.In fact they are not convinced about the utility and have some concerns about safety.
The study as it is however does not add much to what is already known.
The authors should provide the number of influenza-,pneumococcal and COVID-19 infection and disease gravity(eventually also mortality numbers)
in that area.Authors should also ask for the number of infections among the participants.
The conclusions are consistent with the the results of the questions asked in web-based survey (which excludes all those persons who do not use the computer)
The references about studies which are critical about the efficacy of influenza vaccines are not mentioned,but they could help to understand the fact that some of the participants are reluctant to get vaccinated especialy if they are younger than 65.
It is also not clear who informs the General Practioners and how (by the vaccine producers?).
The terms infection and disease are used almost as synonyms.This may explain why younger persons do not belive that they may get infected more often than "normal" persons.
Authors should provide data about this topic.They also should provide data indicating that immunosuppressed persons have a moe severe disease as infected persons with symptoms may get immunosuppressants such as corticosteroid or antibodies aganst acute-phase mediators.

Author Response

The introduction and discussion parts need reshaping.

Authors’ responses : We have reshaped the introduction and discussion in light of Reviewer 1 comments including data on the burden of influenza and pneumococcal diseases in the general population and immunocompromised. We have also discussed the issue of uncertain vaccine effectiveness as an explanation for low vaccine coverage in the discussion, using the suggested references.

The reluctance of people to allow vaccination should be explained.

Authors’ responses: As suggested by the reviewer, we have further discussed arguments that may explain patients’ reluctance for vaccination especially doubts on influenza vaccine effectiveness.

In fact some publications which allow some uncertanity should be mentioned:

Heo JY Human Vaccines and immunotherapeutics 2018;14(3) 744-749,

Okoli GN,Open Forum Infectuous Diseases:Decline in seasonal Influenza Vaccine Effectivness....,Chow EJ,JAMA Network 2020 March 20.

Dänemark has vaccinated 90% of the population against COVID-19 but infection numbers are exploding!

It is also important to discuss why patients seek help in the emergency rooms and possibly die in the hospital as this offers one justification for vaccination

Authors’ responses: As suggested by the reviewer, we have added a sentence in the introduction to highlight the direct and related burden of respiratory infections in adults.

The authors should provide the number of influenza-,pneumococcal and COVID-19 infection and disease gravity(eventually also mortality numbers) in that area.

Authors’ responses: These exact numbers are difficult to find at national or European levels especially in subgroups of patients as the immunocompromised. We had rather add, in the introduction, the best references to our knowledge estimating the overall burden of these diseases and some showing the higher rate of infection and severe forms of the diseases in immunocompromised patients.

Authors should also ask for the number of infections among the participants.

Authors’ responses: Unfortunately, this study was not tailored to assess the number of infections or vaccine efficacy due to its design with a high risk of recall bias and not confirmed declared infection.

The conclusions are consistent with the the results of the questions asked in web-based survey (which excludes all those persons who do not use the computer)

The references about studies which are critical about the efficacy of influenza vaccines are not mentioned, but they could help to understand the fact that some of the participants are reluctant to get vaccinated especialy if they are younger than 65.

Authors’ responses: As requested, we have added a small paragraph highlighting this point using the suggested references. See Discussion 4.3

It is also not clear who informs the General Practioners and how (by the vaccine producers?).

Authors’ responses: General Practitioners are informed on vaccine availability, efficacy/effectiveness, safety, and related recommendations by continuous professional development, scientific congress, national health agencies, and vaccine producers.

They also should provide data indicating that immunosuppressed persons have a moe severe disease as infected persons with symptoms may get immunosuppressants such as corticosteroid or antibodies aganst acute-phase mediators

Authors’ responses: As suggested by the reviewer, we have added references showing the higher rate of infection and severe forms of the diseases in immunocompromised patients.

Reviewer 2 Report

Ethical approval is not reported. Please add.

Setting data are missing. Please, provide.

How was privacy guaranteed? please add.

Table 2, can you explain p value to which test is referring to?

Table 2, how did % are calculated? denominator?

Results section is too long and in most of the time reports the same content of the tables. Please, make it more concise and focus on main results.

Is it possible to combine tables?

line 212 is more consideration compared to results. Move to the discussion.

Remove from the tables the bulleted list.

Be consistent in using COVID or Covid-19.

See errors in lines 287, 288 and 302

I think in the discussion a comparison with data from other European countries can improve the quality of the data interpretation.

Author Response

Ethical approval is not reported. Please add.

Authors’ responses: The ethics committee of our institution considered that this work did not need approval from their side as all data have been gathered from questionnaires and not medical charts. Since May 2019 there is no need as well for CNIL declaration because all studies must follow the EU Data Protection Regulation 2016/679; “GDPR” and e-privacy regulations, which is the case of this work.

Setting data are missing. Please, provide.

Authors’ responses: This study was an online questionnaire of patients all over the country. There has been no specific setting involved.

How was privacy guaranteed? please add.

Authors’ responses: As stated in the ethics considerations section,

  • this study complied with all applicable laws and regulations relating to privacy, security, and data protection (including but not limited to EU Data Protection Regulation 2016/679; “GDPR” and e-privacy regulations); 
  • Ipsos provided appropriate notice to and/or obtained the prior and explicit consent from the individuals to use their data for the provision of Services and creation of the Deliverable including participating in the survey
  • participants were asked to submit an online informed consent form for their participation in the study before accessing the questionnaire. Data were collected anonymously, and participants had the right to access their answers. Only the aggregate data were analyzed and shown in the final report.

Table 2, can you explain p value to which test is referring to?

Authors’ responses: As requested a footnote has been added to Table 2 (Fisher’s exact test)

Table 2, how did % are calculated? denominator?

Authors’ responses: The denominator is the N in each row. The denominator has been added to the first row as an example.

Results section is too long and in most of the time reports the same content of the tables. Please, make it more concise and focus on main results.

Authors’ responses: As suggested by the reviewer, we have shortened the result section to be more concise. Information contained in the tables is not reported in the text.

line 212 is more consideration compared to results. Move to the discussion.

Authors’ responses: We thank the reviewer for this comment. We have deleted this sentence.

Remove from the tables the bulleted list.

Authors’ responses: This change has been done

Be consistent in using COVID or Covid-19.

Authors’ responses: This change has been done

See errors in lines 287, 288 and 302

Authors’ responses: This change has been done

I think in the discussion a comparison with data from other European countries can improve the quality of the data interpretation.

Authors’ responses: As suggested by the reviewer with have further discussed our results in light of European studies. We have added data on immunization rates in at-risk adults from seven European countries from an ECDC report and a recent publication from Germany. We have also added data on factors associated with higher vaccine uptake in Belgium and Spain and pneumococcal immunization rate in the UK and Germany.

Round 2

Reviewer 1 Report

The quality of the manuscript is now better but the paper published by Chow et al. must be introduced and discussed before the manuscript can be published

Author Response

The quality of the manuscript is now better but the paper published by Chow et al. must be introduced and discussed before the manuscript can be published

Authors’ response: We apologize for having missed this reference suggested in the first round of Reviewer 1 comments.

As requested, we have added and discussed this reference in the introduction part.